# Maximal Sparsity with Deep Networks?

**Bo Xin**[1,2]    **Yizhou Wang**[1]    **Wen Gao**[1]    **Baoyuan Wang**[3]    **David Wipf**[2]
[1]Peking University    [2]Microsoft Research, Beijing    [3]Microsoft Research, Redmond
{boxin, baoyuanw, davidwip}@microsoft.com   {yizhou.wang, wgao}@pku.edu.cn

## Abstract

The iterations of many sparse estimation algorithms are comprised of a fixed linear filter cascaded with a thresholding nonlinearity, which collectively resemble a typical neural network layer. Consequently, a lengthy sequence of algorithm iterations can be viewed as a deep network with shared, hand-crafted layer weights. It is therefore quite natural to examine the degree to which a learned network model might act as a viable surrogate for traditional sparse estimation in domains where ample training data is available. While the possibility of a reduced computational budget is readily apparent when a ceiling is imposed on the number of layers, our work primarily focuses on estimation accuracy. In particular, it is well-known that when a signal dictionary has coherent columns, as quantified by a large RIP constant, then most tractable iterative algorithms are unable to find maximally sparse representations. In contrast, we demonstrate both theoretically and empirically the potential for a trained deep network to recover minimal $\ell_0$-norm representations in regimes where existing methods fail. The resulting system, which can effectively learn novel iterative sparse estimation algorithms, is deployed on a practical photometric stereo estimation problem, where the goal is to remove sparse outliers that can disrupt the estimation of surface normals from a 3D scene.

## 1   Introduction

Our launching point is the optimization problem

$$\min_{\boldsymbol{x}} \|\boldsymbol{x}\|_0 \quad \text{s.t. } \boldsymbol{y} = \boldsymbol{\Phi x}, \tag{1}$$

where $\boldsymbol{y} \in \mathbb{R}^n$ is an observed vector, $\boldsymbol{\Phi} \in \mathbb{R}^{n \times m}$ is some known, overcomplete dictionary of feature/basis vectors with $m > n$, and $\|\cdot\|_0$ denotes the $\ell_0$ norm of a vector, or a count of the number of nonzero elements. Consequently, (1) can be viewed as the search for a maximally sparse feasible vector $\boldsymbol{x}^*$ (or approximately feasible if the constraint is relaxed). Unfortunately however, direct assault on (1) involves an intractable, combinatorial optimization process, and therefore efficient alternatives that return a maximally sparse $\boldsymbol{x}^*$ with high probability in restricted regimes are sought. Popular examples with varying degrees of computational overhead include convex relaxations such as $\ell_1$-norm minimization [2, 5, 21], greedy approaches like orthogonal matching pursuit (OMP) [18, 22], and many flavors of iterative hard-thresholding (IHT) [3, 4].

Variants of these algorithms find practical relevance in numerous disparate domains, including feature selection [7, 8], outlier removal [6, 13], compressive sensing [5], and source localization [1, 16]. However, a fundamental weakness underlies them all: *If the Gram matrix $\boldsymbol{\Phi}^\top \boldsymbol{\Phi}$ has significant off-diagonal energy, indicative of strong coherence between columns of $\boldsymbol{\Phi}$, then estimation of $\boldsymbol{x}^*$ may be extremely poor.* Loosely speaking this occurs because, as higher correlation levels are present, the null-space of $\boldsymbol{\Phi}$ is more likely to include large numbers of approximately sparse vectors that tend to distract existing algorithms in the feasible region, an unavoidable nuisance in many practical applications.

In this paper we consider recent developments in the field of deep learning as an entry point for improving the performance of sparse recovery algorithms. Although seemingly unrelated at first

glance, the layers of a deep neural network (DNN) can be viewed as iterations of some algorithm that have been unfolded into a network structure [9, 11]. In particular, iterative thresholding approaches such as IHT mentioned above typically involve an update rule comprised of a fixed, linear filter followed by a non-linear activation function that promotes sparsity. Consequently, algorithm execution can be interpreted as passing an input through an extremely deep network with constant weights (dependent on $\mathbf{\Phi}$) at every layer. This 'unfolding' viewpoint immediately suggests that we consider substituting discriminatively learned weights in place of those inspired by the original sparse recovery algorithm. For example, it has been argued that, given access to a sufficient number of $\{\boldsymbol{x}^*, \boldsymbol{y}\}$ pairs, a trained network may be capable of producing quality sparse estimates with a few number of layers. This in turn can lead to a dramatically reduced computational burden relative to purely optimization-based approaches [9, 19, 23] or to enhanced non-linearities for use with traditional iterative algorithms [15].

While existing empirical results are promising, especially in terms of the reduction in computational footprint, there is as of yet no empirical demonstration of a learned deep network that can unequivocally recover maximally sparse vectors $\boldsymbol{x}^*$ with greater accuracy than conventional, state-of-the-art optimization-based algorithms, especially with a highly coherent $\mathbf{\Phi}$. Nor is there supporting theoretical evidence elucidating the exact mechanism whereby learning may be expected to improve the estimation accuracy, especially in the presence of coherent dictionaries. This paper attempts to fill in some of these gaps, and our contributions can be distilled to the following points:

**Quantifiable Benefits of Unfolding:** We rigorously dissect the benefits of unfolding conventional sparse estimation algorithms to produce trainable deep networks. This includes a precise characterization of exactly how different architecture choices can affect the ability to improve so-called restrictive isometry property (RIP) constants, which measure the degree of disruptive correlation in $\mathbf{\Phi}$. This helps to quantify the limits of shared layer weights, which are the standard template of existing methods [9, 19, 23], and motivates more flexible network constructions reminiscent of LSTM cells [12] that account for multi-resolution structure in $\mathbf{\Phi}$ in a previously unexplored fashion. Note that we defer all proofs, as well as many additional analyses and problem details, to a longer companion paper [26].

**Isolation of Important Factors:** Based on these theoretical insights, and a better understanding of the essential factors governing performance, we establish the degree to which it is favorable to diverge from strict conformity to any particular unfolded algorithmic script. In particular, we argue that layer-wise independent weights and/or activations are essential, while retainment of original thresholding non-linearities and squared-error loss implicit to many sparse algorithms is not. We also recast the the core problem as deep multi-label classification given that optimal support pattern recovery is the primary concern. This allows us to adopt a novel training paradigm that is less sensitive to the specific distribution encountered during testing. Ultimately, we development the first, ultra-fast sparse estimation algorithm (or more precisely a learning procedure that produces such an algorithm) that can effectively deal with coherent dictionaries and adversarial RIP constants.

**State-of-the-Art Empirical Performance:** We apply the proposed system to a practical photometric stereo computer vision problem, where the goal is to estimate the 3D geometry of an object using only 2D photos taken from a single camera under different lighting conditions. In this context, shadows and specularities represent sparse outliers that must be simultaneously removed from $\sim 10^4 - 10^6$ surface points. We achieve state-of-the-art performance using only weak supervision despite a minuscule computational budget appropriate for real-time mobile environments.

## 2 From Iterative Hard Thesholding (IHT) to Deep Neural Networks

Although any number of iterative algorithms could be adopted as our starting point, here we examine IHT because it is representative of many other sparse estimation paradigms and is amenable to theoretical analysis. With knowledge of an upper bound on the true cardinality, solving (1) can be replaced by the equivalent problem

$$\min_{\boldsymbol{x}} \tfrac{1}{2}\|\boldsymbol{y} - \mathbf{\Phi}\boldsymbol{x}\|_2^2 \quad \text{s.t.} \ \|\boldsymbol{x}\|_0 \leq k. \tag{2}$$

IHT attempts to minimize (2) using what can be viewed as computationally-efficient projected gradient iterations [3]. Let $\boldsymbol{x}^{(t)}$ denote the estimate of some maximally sparse $\boldsymbol{x}^*$ after $t$ iterations. The aggregate IHT update computes

$$\boldsymbol{x}^{(t+1)} = H_k \left[ \boldsymbol{x}^{(t)} - \mu \mathbf{\Phi}^\top \left( \mathbf{\Phi}\boldsymbol{x}^{(t)} - \boldsymbol{y} \right) \right], \tag{3}$$

where $\mu$ is a step-size parameter and $H_k[\cdot]$ is a hard-thresholding operator that sets all but the $k$ largest values (in magnitude) of a vector to zero. For the vanilla version of IHT, the step-size $\mu = 1$ leads to a number of recovery guarantees whereby iterating (3), starting from $\boldsymbol{x}^{(0)} = \boldsymbol{0}$ is guaranteed to reduce (2) at each step before eventually converging to the globally optimal solution. These results hinge on properties of $\boldsymbol{\Phi}$ which relate to the coherence structure of dictionary columns as encapsulated by the following definition.

**Definition 1 (Restricted Isometry Property)** *A dictionary $\boldsymbol{\Phi}$ satisfies the Restricted Isometry Property (RIP) with constant $\delta_k[\boldsymbol{\Phi}] < 1$ if*

$$(1 - \delta_k[\boldsymbol{\Phi}])\|\boldsymbol{x}\|_2^2 \leq \|\boldsymbol{\Phi}\boldsymbol{x}\|_2^2 \leq (1 + \delta_k[\boldsymbol{\Phi}])\|\boldsymbol{x}\|_2^2 \tag{4}$$

*holds for all $\{\boldsymbol{x} : \|\boldsymbol{x}\|_0 \leq k\}$.*

In brief, the smaller the value of the RIP constant $\delta_k[\boldsymbol{\Phi}]$, the closer any sub-matrix of $\boldsymbol{\Phi}$ with $k$ columns is to being orthogonal (i.e., it has less correlation structure). It is now well-established that dictionaries with smaller values of $\delta_k[\boldsymbol{\Phi}]$ lead to sparse recovery problems that are inherently easier to solve. For example, in the context of IHT, it has been shown [3] that if $\boldsymbol{y} = \boldsymbol{\Phi}\boldsymbol{x}^*$, with $\|\boldsymbol{x}^*\|_0 \leq k$ and $\delta_{3k}[\boldsymbol{\Phi}] < 1/\sqrt{32}$, then at iteration $t$ of (3) we will have $\|\boldsymbol{x}^{(t)} - \boldsymbol{x}^*\|_2 \leq 2^{-t}\|\boldsymbol{x}^*\|_2$. It follows that as $t \to \infty$, $\boldsymbol{x}^{(t)} \to \boldsymbol{x}^*$, meaning that we recover the true, generating $\boldsymbol{x}^*$. Moreover, it can be shown that this $\boldsymbol{x}^*$ is also the unique, optimal solution to (1) [5].

The success of IHT in recovering maximally sparse solutions crucially depends on the RIP-based condition that $\delta_{3k}[\boldsymbol{\Phi}] < 1/\sqrt{32}$, which heavily constrains the degree of correlation structure in $\boldsymbol{\Phi}$ that can be tolerated. While dictionaries with columns drawn independently and uniformly from the surface of a unit hypersphere (or with elements drawn iid from $\mathcal{N}(0, 1/n)$ ) will satisfy this condition with high probability provided $k$ is small enough [6], for many/most practical problems of interest we cannot rely on this type of IHT recovery guarantee. In fact, except for randomized dictionaries in high dimensions where tight bounds exist, we cannot even compute the value of $\delta_{3k}[\boldsymbol{\Phi}]$, which requires calculating the spectral norm of $\binom{m}{3k}$ subsets of dictionary columns.

There are many ways nature might structure a dictionary such that IHT (or most any other existing sparse estimation algorithm) will fail. Here we examine one of the most straightforward forms of dictionary coherence that can easily disrupt performance. Consider the situation where $\boldsymbol{\Phi} = \left[\epsilon\boldsymbol{A} + \boldsymbol{u}\boldsymbol{v}^\top\right]\boldsymbol{N}$, where columns of $\boldsymbol{A} \in \mathbb{R}^{n \times m}$ and $\boldsymbol{u} \in \mathbb{R}^n$ are drawn iid from the surface of a unit hypersphere, while $\boldsymbol{v} \in \mathbb{R}^m$ is arbitrary. Additionally, $\epsilon > 0$ is a scalar and $\boldsymbol{N}$ is a diagonal normalization matrix that scales each column of $\boldsymbol{\Phi}$ to have unit $\ell_2$ norm. It then follows that if $\epsilon$ is sufficiently small, the rank-one component begins to dominate, and there is no value of $3k$ such that $\delta_{3k}[\boldsymbol{\Phi}] < 1/\sqrt{32}$. In this type of problem we hypothesize that DNNs provide a potential avenue for improvement to the extent that they might be able to compensate for disruptive correlations in $\boldsymbol{\Phi}$.

For example, at the most basic level we might consider general networks with the layer $t$ defined by

$$\boldsymbol{x}^{(t+1)} = f\left[\boldsymbol{\Psi}\boldsymbol{x}^{(t)} + \boldsymbol{\Gamma}\boldsymbol{y}\right], \tag{5}$$

where $f : \mathbb{R}^m \to \mathbb{R}^m$ is a non-linear activation function, and $\boldsymbol{\Psi} \in \mathbb{R}^{m \times m}$ and $\boldsymbol{\Gamma} \in \mathbb{R}^{m \times n}$ are arbitrary. Moreover, given access to training pairs $\{\boldsymbol{x}^*, \boldsymbol{y}\}$, where $\boldsymbol{x}^*$ is a sparse vector such that $\boldsymbol{y} = \boldsymbol{\Phi}\boldsymbol{x}^*$, we can optimize $\boldsymbol{\Psi}$ and $\boldsymbol{\Gamma}$ using traditional stochastic gradient descent just like any other DNN structure. We will first precisely characterize the extent to which this adaptation affords any benefit over IHT where $f(\cdot) = H_k[\cdot]$. Later we will consider flexible, layer-specific non-linearities $f^{(t)}$ and parameters $\{\boldsymbol{\Psi}^{(t)}, \boldsymbol{\Gamma}^{(t)}\}$.

## 3   Analysis of Adaptable Weights and Activations

For simplicity in this section we restrict ourselves to the fixed hard-threshold operator $H_k[\cdot]$ across all layers; however, many of the conclusions borne out of our analysis nonetheless carry over to a much wider range of activation functions $f$. In general it is difficult to analyze how arbitrary $\boldsymbol{\Psi}$ and $\boldsymbol{\Gamma}$ may improve upon the fixed parameterization from (3) where $\boldsymbol{\Psi} = \boldsymbol{I} - \boldsymbol{\Phi}^\top\boldsymbol{\Phi}$ and $\boldsymbol{\Gamma} = \boldsymbol{\Phi}^\top$ (assuming $\mu = 1$). Fortunately though, we can significantly collapse the space of potential weight matrices by including the natural requirement that if $\boldsymbol{x}^*$ represents the true, maximally sparse solution, then it must be a fixed-point of (5). Indeed, without this stipulation the iterations could

diverge away from the globally optimal value of $\boldsymbol{x}$, something IHT itself will never do. These considerations lead to the following:

**Proposition 1** *Consider a generalized IHT-based network layer given by (5) with $f(\cdot) = H_k[\cdot]$ and let $\boldsymbol{x}^*$ denote any unique, maximally sparse feasible solution to $\boldsymbol{y} = \boldsymbol{\Phi}\boldsymbol{x}$ with $\|\boldsymbol{x}\|_0 \leq k$. Then to ensure that any such $\boldsymbol{x}^*$ is a fixed point it must be that $\boldsymbol{\Psi} = \boldsymbol{I} - \boldsymbol{\Gamma}\boldsymbol{\Phi}$.*

Although $\boldsymbol{\Gamma}$ remains unconstrained, this result has restricted $\boldsymbol{\Psi}$ to be a rank-$n$ factor, parameterized by $\boldsymbol{\Gamma}$, subtracted from an identity matrix. Certainly this represents a significant contraction of the space of 'reasonable' parameterizations for a general IHT layer. In light of Proposition 1, we may then further consider whether the added generality of $\boldsymbol{\Gamma}$ (as opposed to the original fixed assignment $\boldsymbol{\Gamma} = \boldsymbol{\Phi}^\top$) affords any further benefit to the revised IHT update

$$\boldsymbol{x}^{(t+1)} = H_k \left[ \left( \boldsymbol{I} - \boldsymbol{\Gamma}\boldsymbol{\Phi} \right) \boldsymbol{x}^{(t)} + \boldsymbol{\Gamma}\boldsymbol{y} \right]. \tag{6}$$

For this purpose we note that (6) can be interpreted as a projected gradient descent step for solving

$$\min_{\boldsymbol{x}} \tfrac{1}{2}\boldsymbol{x}^\top \boldsymbol{\Gamma}\boldsymbol{\Phi}\boldsymbol{x} - \boldsymbol{x}^\top \boldsymbol{\Gamma}\boldsymbol{y} \quad \text{s.t. } \|\boldsymbol{x}\|_0 \leq k. \tag{7}$$

However, if $\boldsymbol{\Gamma}\boldsymbol{\Phi}$ is not positive semi-definite, then this objective is no longer even convex, and combined with the non-convex constraint is likely to produce an even wider constellation of troublesome local minima with no clear affiliation with the global optimum of our original problem from (2). Consequently it does not immediately appear that $\boldsymbol{\Gamma} \neq \boldsymbol{\Phi}^\top$ is likely to provide any tangible benefit. However, there do exist important exceptions. The first indication of how learning a general $\boldsymbol{\Gamma}$ might help comes from the following result:

**Proposition 2** *Suppose that $\boldsymbol{\Gamma} = \boldsymbol{D}\boldsymbol{\Phi}^\top \boldsymbol{W}\boldsymbol{W}^\top$, where $\boldsymbol{W}$ is an arbitrary matrix of appropriate dimension and $\boldsymbol{D}$ is a full-rank diagonal that jointly solve*

$$\delta_{3k}^* [\boldsymbol{\Phi}] \triangleq \inf_{\boldsymbol{W},\boldsymbol{D}} \delta_{3k} [\boldsymbol{W}\boldsymbol{\Phi}\boldsymbol{D}]. \tag{8}$$

*Moreover, assume that $\boldsymbol{\Phi}$ is substituted with $\boldsymbol{\Phi}\boldsymbol{D}$ in (6), meaning we have simply replaced $\boldsymbol{\Phi}$ with a new dictionary that has scaled columns. Given these qualifications, if $\boldsymbol{y} = \boldsymbol{\Phi}\boldsymbol{x}^*$, with $\|\boldsymbol{x}^*\|_0 \leq k$ and $\delta_{3k}^* [\boldsymbol{\Phi}] < 1/\sqrt{32}$, then at iteration $t$ of (6)*

$$\|\boldsymbol{D}^{-1}\boldsymbol{x}^{(t)} - \boldsymbol{D}^{-1}\boldsymbol{x}^*\|_2 \leq 2^{-t}\|\boldsymbol{D}^{-1}\boldsymbol{x}^*\|_2. \tag{9}$$

It follows that as $t \to \infty$, $\boldsymbol{x}^{(t)} \to \boldsymbol{x}^*$, meaning that we recover the true, generating $\boldsymbol{x}^*$. Additionally, it can be guaranteed that after a finite number of iterations, the correct support pattern will be discovered. And it should be emphasized that rescaling $\boldsymbol{\Phi}$ by some known diagonal $\boldsymbol{D}$ is a common prescription for sparse estimation (e.g., column normalization) that does not alter the optimal $\ell_0$-norm support pattern.[1]

But the real advantage over regular IHT comes from the fact that $\delta_{3k}^* [\boldsymbol{\Phi}] \leq \delta_k [\boldsymbol{\Phi}]$, and in many practical cases, $\delta_{3k}^* [\boldsymbol{\Phi}] \ll \delta_{3k} [\boldsymbol{\Phi}]$, which implies success can be guaranteed across a much wider range of RIP conditions. For example, if we revisit the dictionary $\boldsymbol{\Phi} = \left[ \epsilon\boldsymbol{A} + \boldsymbol{u}\boldsymbol{v}^\top \right] \boldsymbol{N}$, an immediate benefit can be observed. More concretely, for $\epsilon$ sufficiently small we argued that $\delta_{3k} [\boldsymbol{\Phi}] > 1/\sqrt{32}$ for all $k$, and consequently convergence to the optimal solution may fail. In contrast, it can be shown that $\delta_{3k}^* [\boldsymbol{\Phi}]$ will remain quite small, satisfying $\delta_{3k}^* [\boldsymbol{\Phi}] \approx \delta_{3k} [\boldsymbol{A}]$, implying that performance will nearly match that of an equivalent recovery problem using $\boldsymbol{A}$ (and as we discussed above, $\delta_{3k} [\boldsymbol{A}]$ is likely to be relatively small per its unique, randomized design). The following result generalizes a sufficient regime whereby this is possible:

**Corollary 1** *Suppose $\boldsymbol{\Phi} = \left[ \epsilon\boldsymbol{A} + \boldsymbol{\Delta}_r \right] \boldsymbol{N}$, where elements of $\boldsymbol{A}$ are drawn iid from $\mathcal{N}(0, 1/n)$, $\boldsymbol{\Delta}_r$ is any arbitrary matrix with $\text{rank}[\boldsymbol{\Delta}_r] = r < n$, and $\boldsymbol{N}$ is a diagonal matrix (e.g, one that enforces unit $\ell_2$ column norms). Then*

$$E \left( \delta_{3k}^* [\boldsymbol{\Phi}] \right) \leq E \left( \delta_{3k} \left[ \widetilde{\boldsymbol{A}} \right] \right), \tag{10}$$

*where $\widetilde{\boldsymbol{A}}$ denotes the matrix $\boldsymbol{A}$ with any $r$ rows removed.*

Additionally, as the size of $\mathbf{\Phi}$ grows proportionally larger, it can be shown that with overwhelming probability $\delta_{3k}^*[\mathbf{\Phi}] \leq \delta_{3k}\left[\widetilde{\mathbf{A}}\right]$. Overall, these results suggest that we can essentially annihilate any potentially disruptive rank-$r$ component $\mathbf{\Delta}_r$ at the cost of implicitly losing $r$ measurements (linearly independent rows of $\mathbf{A}$, and implicitly the corresponding elements of $\mathbf{y}$). Therefore, at least provided that $r$ is sufficiently small such that $\delta_{3k}\left[\widetilde{\mathbf{A}}\right] \approx \delta_{3k}[\mathbf{A}]$, we can indeed be confident that a modified form of IHT can perform much like a system with an ideal RIP constant. And of course in practice we may not know how $\mathbf{\Phi}$ decomposes as some $\mathbf{\Phi} \approx [\epsilon\mathbf{A} + \mathbf{\Delta}_r]\mathbf{N}$; however, to the extent that this approximation can possibly hold, the RIP constant can be improved nonetheless.

It should be noted that globally solving (8) is non-differentiable and intractable, but this is the whole point of incorporating a DNN network to begin with. If we have access to a large number of training pairs $\{\mathbf{x}^*, \mathbf{y}\}$ generated using the true $\mathbf{\Phi}$, then during the course of the learning process a useful $\mathbf{W}$ and $\mathbf{D}$ can be implicitly estimated such that a maximal number of sparse vectors can be successfully recovered. Of course we will experience diminishing marginal returns as more non-ideal components enter the picture. In fact, it is not difficult to describe a slightly more sophisticated scenario such that use of layer-wise constant weights and activations are no longer capable of lowering $\delta_{3k}[\mathbf{\Phi}]$ significantly at all, portending failure when it comes to accurate sparse recovery.

One such example is a *clustered dictionary model* (which we describe in detail in [26]), whereby columns of $\mathbf{\Phi}$ are grouped into a number of tight clusters with minimal angular dispersion. While the clusters themselves may be well-separated, the correlation *within* clusters can be arbitrarily large. In some sense this model represents the simplest partitioning of dictionary column correlation structure into two scales: the inter- and intra-cluster structures. Assuming the number of such clusters is larger than $n$, then layer-wise constant weights and activations are unlikely to provide adequate relief, since the implicit $\mathbf{\Delta}_r$ factor described above will be full rank.

Fortunately, simple adaptations of IHT, which are reflective of many generic DNN structures, can remedy the problem. The core principle is to design a network such that earlier layers/iterations are tasked with exposing the correct support at the cluster level, without concern for accuracy within each cluster. Once the correct cluster support has been obtained, later layers can then be charged with estimating the fine-grain details of within-cluster support. We believe this type of multi-resolution sparse estimation is essential when dealing with highly coherent dictionaries. This can be accomplished with the following adaptations to IHT:

1. The hard-thresholding operator is generalized to 'remember' previously learned cluster-level sparsity patterns, in much the same way that LSTM gates allow long term dependencies to propagate [12] or highway networks [20] facilitate information flow unfettered to deeper layers. Practically speaking this adaptation can be computed by passing the prior layer's activations $\mathbf{x}^{(t)}$ through linear filters followed by indicator functions, again reminiscent of how DNN gating functions are typically implemented.

2. We allow the layer weights $\{\mathbf{\Psi}^{(t)}, \mathbf{\Gamma}^{(t)}\}$ to vary from iteration to iteration $t$ sequencing through a fixed set akin to layers of a DNN.

In [26] we show that hand-crafted versions of these changes allow IHT to provably recovery maximally sparse vectors $\mathbf{x}^*$ in situations where existing algorithms fail.

## 4 Discriminative Multi-Resolution Sparse Estimation

As implied previously, guaranteed success for most existing sparse estimation strategies hinges on the dictionary $\mathbf{\Phi}$ having columns drawn (approximately) from a uniform distribution on the surface of a unit hypersphere, or some similar condition to ensure that subsets of columns behave approximately like an orthogonal basis. Essentially this confines the structure of the dictionary to operate on a single universal scale. The clustered dictionary model described in the previous section considers a dictionary built on two different scales, with a cluster-level distribution (coarse) and tightly-packed within-cluster details (fine). But practical dictionaries may display structure operating across a variety of scales that interleave with one another, forming a continuum among multiple levels.

When the scales are clearly demarcated, we have argued that it is possible to manually define a multi-resolution IHT-inspired algorithm that guarantees success in recovering the optimal support pattern; and indeed, IHT could be extended to handle a clustered dictionary model with nested

structures across more than two scales. However, without clearly partitioned scales it is much less obvious how one would devise an optimal IHT modification. It is in this context that learning flexible algorithm iterations is likely to be most advantageous. In fact, the situation is not at all unlike many computer vision scenarios whereby handcrafted features such as SIFT may work optimally in confined, idealized domains, while learned CNN-based features are often more effective otherwise.

Given a sufficient corpus of $\{\boldsymbol{x}^*, \boldsymbol{y}\}$ pairs linked via some fixed $\boldsymbol{\Phi}$, we can replace manual filter construction with a learning-based approach. On this point, although we view our results from Section 3 as a convincing proof of concept, it is unlikely that there is anything intrinsically special about the specific hard-threshold operator and layer-wise construction we employed per se, as long as we allow for deep, adaptable layers that can account for structure at multiple scales. For example, we expect that it is more important to establish a robust training pipeline that avoids stalling at the hand of vanishing gradients in a deep network, than to preserve the original IHT template analogous to existing learning-based methods. It is here that we propose several deviations:

**Multi-Label Classification Loss:** We exploit the fact that in producing a maximally sparse vector $\boldsymbol{x}^*$, the main challenge is estimating supp$[\boldsymbol{x}^*]$. Once the support is obtained, computing the actual nonzero coefficients just boils down to solving a least squares problem. But any learning system will be unaware of this and could easily expend undue effort in attempting to match coefficient magnitudes at the expense of support recovery. Certainly the use of a data fit penalty of the form $\|\boldsymbol{y} - \boldsymbol{\Phi}\boldsymbol{x}\|_2^2$, as is adopted by nearly all sparse recovery algorithms, will expose us to this issue. Therefore we instead formulate sparse recovery as a multi-label classification problem. More specifically, instead of directly estimating $\boldsymbol{x}^*$, we attempt to learn $\boldsymbol{s}^* = [s_1^*, \ldots, s_m^*]^\top$, where $s_i^*$ equals the indicator function $\mathbb{I}[x_i^* \neq 0]$. For this purpose we may then incorporate a traditional multi-label classification loss function via a final softmax output layer, which forces the network to only concern itself with learning support patterns. This substitution is further justified by the fact that even with traditional IHT, the support pattern will be accurately recovered before the iterations converge exactly to $\boldsymbol{x}^*$. Therefore we may expect that fewer layers (as well as training data) are required if all we seek is a support estimate, opening the door for weaker forms of supervision.

**Instruments for Avoiding Bad Local Solutions:** Given that IHT can take many iterations to converge on challenging problems, we may expect that a relatively deep network structure will be needed to obtain exact support recovery. We must therefore take care to avoid premature convergence to local minima or areas with vanishing gradient by incorporating several recent countermeasures proposed in the DNN community. For example, the adaptive variant of IHT described previously is reminiscent of highway networks or LSTM cells, which have been proposed to allow longer range flow of gradient information to improve convergence through the use of gating functions. An even simpler version of this concept involves direct, un-gated connections that allow much deeper 'residual' networks to be trained [10] (which is even suggestive of the residual factor embedded in the original IHT iterations). We deploy this tool, along with batch-normalization [14] to aid convergence, for our basic feedforward pipeline, along with an alternative structure based on recurrent LSTM cells. Note that unfolded LSTM networks frequently receive a novel input for every time step, whereas here $\boldsymbol{y}$ is applied unaltered at every layer (more on this in [26]). We also replace the non-integrable hard-threshold operator with simple rectilinear (ReLu) units [17], which are functionally equivalent to one-sided soft-thresholding; this convex selection likely reduces the constellation of sub-optimal local minima during the training process.

## 5  Experiments and Applications

**Synthetic Tests with Correlated Dictionaries:** We generate a dictionary matrix $\boldsymbol{\Phi} \in \mathbb{R}^{n \times m}$ using $\boldsymbol{\Phi} = \sum_{i=1}^n \frac{1}{i^2} \boldsymbol{u}_i \boldsymbol{v}_i^\top$, where $\boldsymbol{u}_i \in \mathbb{R}^n$ and $\boldsymbol{v}_i \in \mathbb{R}^m$ have iid elements drawn from $\mathcal{N}(0, 1)$. We also rescale each column of $\boldsymbol{\Phi}$ to have unit $\ell_2$ norm. $\boldsymbol{\Phi}$ generated in this way has super-linear decaying singular values (indicating correlation between the columns) but is not constrained to any specific structure. Many dictionaries in real applications have such a property. As a basic experiment, we generate $N = 700000$ ground truth samples $\boldsymbol{x}^* \in \mathbb{R}^m$ by randomly selecting $d$ nonzero entries, with nonzero amplitudes drawn iid from the uniform distribution $\mathcal{U}[-0.5, 0.5]$, excluding the interval $[-0.1, 0.1]$ to avoid small, relatively inconsequential contributions to the support pattern. We then create $\boldsymbol{y} \in \mathbb{R}^n$ via $\boldsymbol{y} = \boldsymbol{\Phi}\boldsymbol{x}^*$. As $d$ increases, the estimation problem becomes more difficult. In fact, to guarantee success with such correlated data (and high RIP constant) requires evaluating on the order of $\binom{m}{n}$ linear systems of size $n \times n$, which is infeasible even for small values, indicative of how challenging it can be to solve sparse inverse problems of any size. We set $n=20$ and $m=100$.

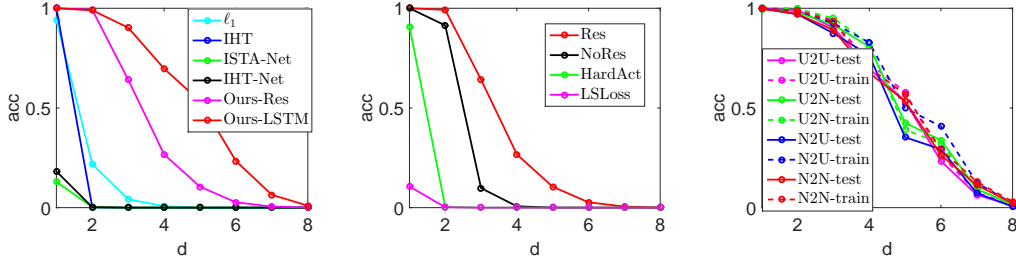

Figure 1: Average support recovery accuracy. Left: Uniformly distributed nonzero elements. Mid: Different network variants. Right: Different training and testing distr. (LSTM-Net results).

We used $N_1 = 600000$ samples for training and the remaining $N_2 = 100000$ for testing. Echoing our arguments in Section 4, we explored both a feedforward network with residual connections [10] and a recurrent network with vanilla LSTM cells [12]. To evaluate the performance, we check whether the $d$ ground truth nonzeros are aligned with the predicted top-$d$ values produced by our network, a common all-or-nothing metric in the compressive sensing literature. Detailed network design, optimization setup, and alternative metrics can be found in [26].

Figure 1(*left*) shows comparisons against a battery of existing algorithms, both learning- and optimization-based. These include standard $\ell_1$ minimization via ISTA iterations [2], IHT [3] (supplied with the ground truth number of nonzeros), an ISTA-based network [9], and an IHT-inspired network [23]. For both the ISTA- and IHT-based networks, we used the exact same training data described above. Note that given the correlated $\mathbf{\Phi}$ matrix, the recovery performance of IHT, and to a lesser degree $\ell_l$ minimization using ISTA, is rather modest as expected given that the associated RIP constant will be quite large by construction. In contrast our two methods achieve uniformly higher accuracy, including over other learning-based methods trained with the same data. This improvement is likely the result of three significant factors: (i) Existing learning methods initialize using weights derived from the original sparse estimation algorithms, but such an initialization will be associated with locally optimal solutions in most cases with correlated dictionaries. (ii) As described in Section 3, constant weights across layers have limited capacity to unravel multi-resolution dictionary structure, especially one that is not confined to only possess some low rank correlating component. (iii) The quadratic loss function used by existing methods does not adequately focus resources on the crux of the problem, which is accurate support recovery. In contrast we adopt an initialization motivated by DNN-based training considerations, unique layer weights to handle a multi-resolution dictionary, and a multi-label classification output layer to focus on support recovery.

To further isolate essential factors affecting performance, we next consider the following changes: (1) We remove the residual connections from Res-Net. (2) We replace ReLU with hard-threshold activations. In particular, we utilize the so-called $\text{HELU}_\sigma$ function introduced in [23], which is a continuous and piecewise linear approximation of the scalar hard-threshold operator. (3) We use a quadratic penalty layer instead of a multi-label classification loss layer, i.e., the loss function is changed to $\sum_{i=1}^{N_1} \|\boldsymbol{a}^{(i)} - \boldsymbol{y}^{(i)}\|_2^2$ (where $\boldsymbol{a}$ is the output of the last fully-connected layer) during training. Figure 1(*middle*) displays the associated recovery percentages, where we observe that in each case performance degrades. Without the residual design, and also with the inclusion of a rigid, non-convex hard-threshold operator, local minima during training appear to be a likely culprit, consistent with observations from [10]. Likewise, use of a least-squares loss function is likely to over-emphasize the estimation of coefficient amplitudes rather than focusing on support recovery.

Finally, from a practical standpoint we may expect that the true amplitude distribution may deviate at times from the original training set. To explore robustness to such mismatch, as well as different amplitude distributions, we consider two sets of candidate data: the original data, and similarly-generated data but with the uniform distribution of nonzero elements replaced with the Gaussians $\mathcal{N}(\pm 0.3, 0.1)$, where the mean is selected with equal probability as either $-0.3$ or $0.3$, thus avoiding tiny magnitudes with high probability. Figure 1(*right*) reports accuracies under different distributions for both training and testing, including mismatched cases. (The results are obtained using LSTM-Net, but the Res-net showed similar pattern.) The label 'U2U' refers to training and testing with the uniformly distributed amplitudes, while 'U2N' uses uniform training set and a Gaussian test set. Analogous definitions apply for 'N2N' and 'N2U'. In all cases we note that the performance is

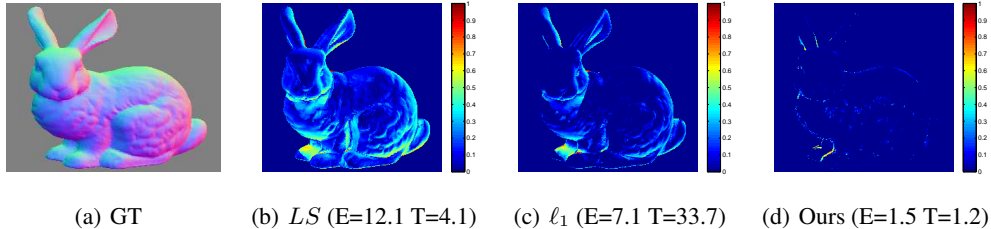

| (a) GT | (b) $LS$ (E=12.1 T=4.1) | (c) $\ell_1$ (E=7.1 T=33.7) | (d) Ours (E=1.5 T=1.2) |

Figure 2: Reconstruction error maps. Angular error in degrees (E) and runtime in sec. (T) are provided.

quite stable across training and testing conditions. We would argue that our recasting of the problem as multi-label classification contributes, at least in part, to this robustness. The application example described next demonstrates further tolerance of training-testing set mismatches.

**Practical Application - Photometric Stereo:** Suppose we have $q$ observations of a given surface point from a Lambertian scene under different lighting directions. Then the resulting measurements from a standard calibrated photometric stereo design (linear camera response function, an orthographic camera projection, and known directional light sources), denoted $o \in \mathbb{R}^q$, can be expressed as $o = \rho Ln$, where $n \in \mathbb{R}^3$ denotes the true 3D surface normal, each row of $L \in \mathbb{R}^{q \times 3}$ defines a lighting direction, and $\rho$ is the diffuse albedo, acting here as a scalar multiplier [24]. If specular highlights, shadows, or other gross outliers are present, then the observations are more realistically modeled as $o = \rho Ln + e$, where $e$ is an an unknown sparse vector [13, 25]. It is apparent that, since $n$ is unconstrained, $e$ need not compensate for any component of $o$ in the range of $L$. Given that $\text{null}[L^\top]$ is the orthogonal complement to $\text{range}[L]$, we may consider the following problem

$$\min_{e} \|e\|_0 \quad \text{s.t. Proj}_{\text{null}[L^\top]}(o) = \text{Proj}_{\text{null}[L^\top]}(e) \tag{11}$$

which ultimately collapses to our canonical sparse estimation problem from (1), where lighting-hardware-dependent correlations may be unavoidable in the implicit dictionary.

Following [13], we use 32-bit HDR gray-scale images of the object *Bunny* ($256 \times 256$) with foreground masks under different lighting conditions whose directions, or rows of $L$, are randomly selected from a hemisphere with the object placed at the center. To apply our method, we first compute $\Phi$ using the appropriate projection operator derived from the lighting matrix $L$. As real-world training data is expensive to acquire, we instead use weak supervision by synthetically generating a training set as follows. First, we draw a support pattern for $e$ randomly with cardinality $d$ sampled uniformly from the range $[d_1, d_2]$. The values of $d_1$ and $d_2$ can be tuned in practice. Nonzero values of $e$ are assigned iid random values from a Gaussian distribution whose mean and variance are also tunable. Beyond this, no attempt was made to match the true outlier distributions encountered in applications of photometric stereo. Finally, for each $e$ we can naturally compute observations via the linear constraint in (11), which serve as candidate network inputs.

Given synthetic training data acquired in this way, we learn a network with the exact same structure and optimization parameters as in Section 5; no application-specific tuning was introduced. We then deploy the resulting network on the gray-scale Bunny images. For each surface point, we use our DNN model to approximately solve (11). Since the network output will be a probability map for the outlier support set instead of the actual values of $\mathbf{e}$, we choose the 4 indices with the *least* probability as inliers and use them to compute $n$ via least squares.

We compare our method against the baseline least squares estimate from [24] and $\ell_1$ norm minimization. We defer more quantitative comparisons to [26]. In Figure 2, we illustrate the recovered surface normal error maps of the hardest case (fewest lighting directions). Here we observe that our DNN estimates lead to far fewer regions of significant error and the runtime is orders of magnitude faster. Overall though, this application example illustrates that weak supervision with mismatched synthetic training data can, at least for some problem domains, be sufficient to learn a quite useful sparse estimation DNN; here one that facilitates real-time 3D modeling in mobile environments.

**Discussion:** In this paper we have shown that deep networks with hand-crafted, multi-resolution structure can provably solve certain specific classes of sparse recovery problems where existing algorithms fail. However, much like CNN-based features can often outperform SIFT on many computer vision tasks, we argue that a discriminative approach can outperform manual structuring of layers/iterations and compensate for dictionary coherence under more general conditions.

**Acknowledgements:** This work was done while the first author was an intern at Microsoft Research, Beijing. It is also funded by 973-2015CB351800, NSFC-61231010, NSFC-61527804, NSFC-61421062, NSFC-61210005 and MOEMicrosoft Key Laboratory, Peking University.

## Footnotes

[1] Inclusion of this diagonal factor $\boldsymbol{D}$ can be equivalently viewed as relaxing Proposition 1 to hold under some fixed rescaling of $\boldsymbol{\Phi}$, i.e., an operation that preserves the optimal support pattern.

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
