[Reviews · NeurIPS 2016]

Reviewer 1

Summary

The authors show that deep networks with hand-crafted structure inspired from IHT can solve sparse recovery problems, in particular with coherent dictionaries and adversarial RIP constants.

Qualitative Assessment

Notes on technical quality: The arguments and claim are very broad while the practical experiments are highly limited. The work deserves more rigorous real data experimental analysis, specially studying multiple contributing factors in isolation. Notes on novelty: The paper has interesting theoretical contributions and implications for some applications that off-the-shelf sparse recovery algorithms may fail. Notes on potential impact: Unfolding conventional sparse estimation algorithms to produce deep networks is interesting. In particular, developing a modified IHT that can perform like a system with an ideal RIP constant can potentially impact derivation of other traditional iterative sparse recovery algorithms so that they can be adapted to the adversarial condition, and better exploit abundance of training data. Notes on qualitative assessment: This is an interesting study, and the authors address many aspects in reasonable depth. However, the paper as a single document is not self-contained; many arguments are only speculatively asserted. This is evidently a small subset of a comprehensive work, but the amount of squeezed information and lack of supporting evidence makes it hard to follow at various parts. More specific remarks: - L4: “... hand-crafted layer weights”: The argument holds in scenarios that the forward model for (natural) dictionaries is presumed or known. How does this generalize to data-driven dictionary learning for sparse representation? - L206: The adaptation to IHT would benefit from precise formalism or block diagrams describing and contrasting multiple network structures. - L241: Similar to above argument, the multi-label classification loss function is not clear. Minor typos: L69: “...the the...” L72: “...we development the first...” L102 & 158: “...we recovery the true...” L155: D should be boldface. Equation 10: define E.

Confidence in this Review

2-Confident (read it all; understood it all reasonably well)


Reviewer 2

Summary

This paper considers the problem of sparse linear estimation with structured dictionary. It shows that trained deep neural network provides better estimation than traditional sparse estimation algorithms (including those using deep learning, [11]) that assume incoherence of the dictionary. They illustrate this for dictionary having low-rank (or approximately low-rank) structure. The main claims are empirical, though some side theoretical insights are provided. The authors stress the importance of layer-wise independent weights.

Qualitative Assessment

On the one hand, sparse linear estimation with structured dictionaries is an important open problem. Attacking it with trained deep neural nets is a nice idea that should be of interest to the NIPS community. On the other hand, for low-rank dictionaries it is not very surprising that multilayer neural network work well, because being low-rank is a kind of two-layer structure. For this reason could maybe even two-layer architecture be sufficient for this task? Although the authors consider some variants of the neural network (changing the non-linearity of the loss function), they do not consider changing the number of layers nor their width. I am also a little reluctant about authors focus on the RIP property, that is sufficient but not necessary for success. Spell checking is still needed.

Confidence in this Review

2-Confident (read it all; understood it all reasonably well)


Reviewer 3

Summary

This manuscript presents a generic deep neural network (DNN) solution for the l_0 norm sparse coding problem. The authors uses Iterative Hard Thresholding (IHT) as an example to justify the rightness of an alternative DNN method, including, hand-crafted and a learning-based models. Computational experiments on synthetic data and real data show better results than existing models such as IHT and l_1 norm sparse coding.

Qualitative Assessment

Overall, this paper proposes a theoretically sound solution for the recovery of l_0 norm sparsity. Since the derivation of this framework is quite general, more potential models can be designed with the inspiration of this work. This work will be quite interesting for reminding machine learning scientists of thinking of deep learning methods for problems hardly solvable by shallow models. The presentation of this manuscript could be improved by providing a figure in the main manuscript showing the structures of the derived DNNs (somehow provided in the supplementary file). Minors: 1. Line 69: the the -> the 2. Line 80: Thesholding -> Thresholding 3. Line 102: recovery -> recover 4. Could not understand line 112. 5. Line 155: matrix D should be in bold face. 6. Line 338: an an -> an 7: Line 367: Discussion/Conclusion should be in a level-1 section.

Confidence in this Review

2-Confident (read it all; understood it all reasonably well)


Reviewer 4

Summary

The paper describes an interpretation of unfolded iterative hard thresholding as deep neural networks. The novelty lies in the use of a multiclass loss instead the squared loss used normally, and the use of batch normalization along LSTM-like cells to avoid bad local minima.

Qualitative Assessment

The paper is well written according to my personal opinion. The novelties introduced, although not being groundbreaking, are effective and interesting improvements over [25]. The improvements on running time and error are notorious.

Confidence in this Review

2-Confident (read it all; understood it all reasonably well)


Reviewer 5

Summary

The paper describes a deep neural network for sparse approximation that deals with dictionary with high RIP constant. The network is inspired by unfolding the operations in the Iterative Hard Thresholding (IHT) algorithm. It is further generalized with a parameterization scheme such that the optimal solution is a fixed point. Compared to the IHT algorithm, it is shown that a network with learnable parameter could potentially find the optimal solution for some dictionaries with high RIP constant. Some structural variation of the neural net are discussed, e.g. softmax loss on the support of the solution and other techniques to avoid local minima. Experiments show that the proposed network outperform the existing algorithms. The authors also discuss the influence of various components of the network on the performance. The algorithm is applied on the task of photometric stereo problem and achieved state-of-the-art result.

Qualitative Assessment

The paper presents an example of a dictionary with high RIP constant, i.e. a matrix with columns randomly draw from a unit ball plus a low rank matrix, and show in proposition 2 that the optimal solution of such dictionary can be found if the parameter in the network is set to a certain value. My concerns are: 1. does the dictionary in the real world application has the same structure as the example used to justified the algorithm? 2. is the parameter learnt by gradient descent the same as or close to the theoretical result in proposition 2? I believe more analysis can be done to strengthen the connection between the empirical success of the algorithm and the theory that motivates it.

Confidence in this Review

1-Less confident (might not have understood significant parts)


Reviewer 6

Summary

This paper investigates the problem of sparse recovery of a linear system with unknown dictionary. Given a set of training signal-response pairs generated from an unknown linear system, the authors introduce a predictive sparse recovery model based on deep neural network (DNN) training. The main idea is to assume that the sparse signal can be encoded by a deep network built over the observation. Inspired by iterative hard-thresholding (IHT), the hard-thresholding operation is applied as activation function on the hidden layers to induce the nonlinearity. Numerical results on synthetic data and photometric stereo applications are reported to show the actual performance of the proposed model.

Qualitative Assessment

The idea of using DNN to approximately unfold the IHT iteration procedure is interesting and novel as far as I am aware of. I have, however, the following major concerns on its content: (1) Motivation is unconvincing: The training pairs (x, y) are assumed to obey a linear system y=Ax with unknown measurement matrix A. Given ample training data, why should we bother to use DNN-type models if A can simply be estimated via, e.g., least squared regression? Apparently, when A is recovered we may then apply IHT on a testing response vector with the estimated measurement matrix to approximately recover the sparse signal. Actually, I note the simulation study uses a training set of size 600,000 and the measurement matrix is only of size 20x100 << 600,000. In such a setting, it should not be uneasy to accurately estimate A from the data. (2) Another major concern is its readability. The somewhat overcomplicated writing style in fact underscores the limited innovation of model and algorithmic contributions. Throughout the paper, I cannot find a clear description of the network structure and its optimization algorithm. (3) There is a clear gap between theory and applications. The main sparse recovery result is about parameter estimation error in a noiseless linear system. The presented application of multi-resolution sparse estimation is defined as a multi-label classification problem to which it remains unclear whether the developed theory can apply.

Confidence in this Review

1-Less confident (might not have understood significant parts)